# Frugal neural reranking: evaluation on the Covid-19 literature

**Tiago Almeida**
IEETA
University of Aveiro
3810-193 Aveiro, Portugal
tiagomeloalmeida@ua.pt

**Sérgio Matos**
DETI/IEETA
University of Aveiro
3810-193 Aveiro, Portugal
aleixomatos@ua.pt

## Abstract

The Covid-19 pandemic urged the scientific community to join efforts at an unprecedented scale, leading to faster than ever dissemination of data and results, which in turn motivated more research works. This paper presents and discusses information retrieval models aimed at addressing the challenge of searching the large number of publications that stem from these studies.

The model presented, based on classical baselines followed by an interaction based neural ranking model, was evaluated and evolved within the TREC Covid challenge setting. Results on this dataset show that, when starting with a strong baseline, our light neural ranking model can achieve results that are comparable to other model architectures that use very large number of parameters.

## 1 Introduction

The emerging pandemic of Covid-19 caused a surge of worldwide scientific studies to be published as a form of open or peer-reviewed articles, as observable in Figure 1. Unfortunately, this growing collection of scientific studies tends to be extremely broad, comprehending diverse presentations of the subject, with different perspectives coming from different scientific areas.

As a consequence, it is continually more challenging to successfully navigate through the amount of information already published about Covid-19, which deprecates precious researching time. So, it is imperative to study and provide ways of successfully navigating (searching) this unstructured type of information (articles), helping the researchers to rapidly find consistent information about their research topic.

This work presents our approach to rapidly address the enunciated search problem over the Covid-19 literature. We made an adaptation of a system developed for document and snippet retrieval that had been evaluated through the BioASQ 8b competition. As a quick reference, BioASQ (Tsatsaronis et al., 2015) provides annual competitions on biomedical semantic indexing and question-answering. So, our hypothesis was to adapt our working BioASQ system (Almeida and Matos, 2020a) to the Covid-19 literature, supported by the assumption that the underlying searching data may be from a close information source, and thus, sharing some similarity.

Our system follows a traditional retrieval pipeline, where an initial search is performed with an efficient retrieval solution to reduce the search space, and then a neural model is leveraged to rerank the articles in this reduced space. However, contrary to what is currently the state-of-the-art in NLP related tasks, we did not follow the trend of using a transform-based architecture for our reranking model. Instead, we built upon an interaction-based model, which yields a final model with only 620 trainable parameters, easing the deployment of a system powered by this pipeline to the public use. Furthermore, although the gains in performance obtained from transform-based models are undeniable when correctly finetuned, it may not be trivial how these can be adapted and applied on unseen data, which corresponds to this special case.

Finally, we built a search engine that was made available[1] on 31st of March[2], and was, to the best of our knowledge, one of the first tools available for public use. Subsequently, other tools have appeared, namely Neural Covidex [3], CORD-19 Search [4], CovidAsk [5], and notably a contribution

---

[1] http://covidsearch.web.ua.pt/
[2] https://twitter.com/IEETA_Research/status/1245009877656879104
[3] http://covidex.ai/
[4] https://cord19.vespa.ai/
[5] https://covidask.korea.ac.kr/

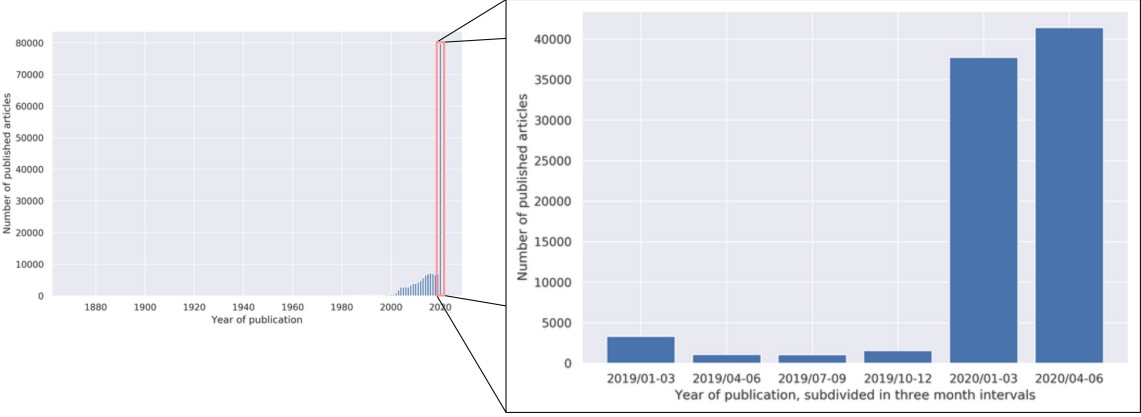

Figure 1: On the left side, it is presented a histogram of the number of articles published per year that are related to the novel coronavirus. On the right side, it is shown the same information over the period 2019/01 to 2020/06 in intervals of three months. The presented information was extracted from the CORD-19 collection

from Google, Covid-19 Research Explorer [6].

Following this section, we show a more detailed description of the retrieval pipeline and present an evaluation of the system on the ongoing competition TREC Covid followed by a thorough analysis/discussion.

## 2 Retrieval Pipeline

To keep this paper self-contained, we will now briefly address our BioASQ 8b system. As previously mentioned, and as shown in Figure 2, we split our pipeline into two phases. First, a traditional IR approach is adopted to search a given collection of documents and retrieve the top **N** documents that share some relevance signal with a specific question. Then, these top documents are reranked using a neural model that explores in more detail the context where the original relevance matches occurred. The following subsections addresses, in more detail, the document collection used and both retrieval phases.

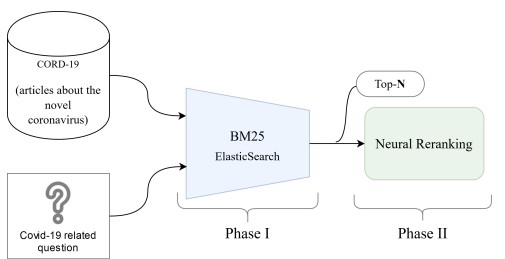

Figure 2: An overview of the pipeline

[6]https://covid19-research-explorer.appspot.com/

### 2.1 Novel coronavirus document collection

An important part of any search engine is the definition of the document collection to be searched. In this case, the document collection is provided by the Allen Institute for Artificial Intelligence (AI2), that released, on March 16, the **Cord-19** open collection of scientific articles about the novel coronavirus. Currently, this collection is updated on a daily basis and has more than 130k articles gathered from peer-reviewed publications and open archives such as bioRxiv and medRxiv. Since the articles are collected from different sources and then converted to a uniform representation, this creates a heterogeneous collection with missing fields for some articles. To overcome this representation problem, we decided to only consider the title and abstract fields, neglecting other fields such as the rest of the document (full-text). There were two main reasons behind this decision. In the first place, scientific abstracts are generally well written and structured, making them well suited as a unit of information to be searched. The second reason concerns our hypothesis of adapting our BioASQ system to this literature. Since only the title and abstract are available in the BioASQ competition, this makes it a closer and more similar task.

### 2.2 Phase-I: Document filter

The first phase of our pipeline has the objective of filtering the continuously growing collection of articles by selecting only the **top-N** potential relevant documents for a given question. Since in this stage the entire collection is searched, it is more important to consider an efficient and scalable solution. With this in mind, we decided to rely on

Elasticsearch (ES), an industry level solution, to index the collection, and used the BM25 (Robertson and Zaragoza, 2009) weighting scheme for the retrieval.

Alternatively, the Anserini toolkit (Yang et al., 2017) has gained increasing interest from the IR community, given the strong baselines that have been achieved, especially on TREC Covid as noticeable in the Results section. Despite this trend, we decided to continue with ES, since it was the same used for the BioASQ task, where it handles a collection of approximately 20 million documents, giving us the confidence that it should be able to easily handle this new collection. Furthermore, we also adopted the same values for the $k1$ and $b$ parameters, which were finetuned for the BioASQ challenge.

## 2.3 Phase-II: Neural ranking model

In the second phase, the previously retrieved **top-N** documents are reranked by a neural model. The rationale here is to consider more and different matching signals comparatively to the previous step, in order to produce the final ranking order. In a more detailed way, the previous step only considers the exact matching signals, i.e., only the words that appear both on the query and the document are taken into account and weighted to produce the phase-I ranking.

The adopted neural ranking model is inspired by the DeepRank (Pang et al., 2017) architecture and represents an enhancement of our previous work (Almeida and Matos, 2020b), with the following major differences: the passage position input, proposed on the original work, was dropped; the detection network and the measure network were simplified and now form the interaction network; the contributions of each passage to the final document score are now assumed to be independent, and hence the self-attention layer proposed in (Almeida and Matos, 2020b) was replaced. The updated architecture can be visualized in Figure 3. Furthermore, the intuition behind this model is to make a thorough evaluation of the article passages where the exact matches occur, by taking into consideration their context. In other words, this model explores the interactions presented in the entire passage of each exact match and makes a more refined judgment of the passage relevance based on that.

Before addressing, in more detail, the proposed model, we first define a **query** as a sequence of

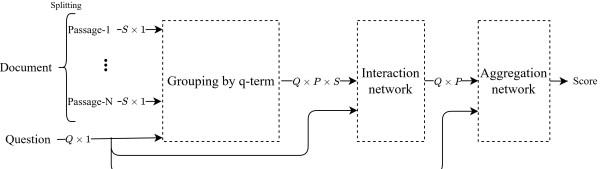

Figure 3: Overview of the neural ranking model with a tensor representation of the data flow.

terms $q = \{u_0, u_1, ..., u_Q\}$, where $u_i$ is the $i$-th term of the query and $Q$ the size of the query; a **document passage** as $p = \{v_0, v_1, ..., v_T\}$, where $v_k$ is the $k$-th term of the passage and $T$ the size of the passage; and a set of document passages as sequence of passages $D = \{p_0, p_1, ..., p_N\}$. A pretrained word2vec model is used for representing query and document terms.

Examining, from left to right, the architecture in Figure 3, the document is first split into a sequence of passages using the nltk.PunktSentenceTokenizer. Then, in the **grouping by q-term** block, each passage is associated to each query term, by verifying if that query term appears in the passage. In other words, this stage created a set of document passages aggregated by each query term as $D(u_i) = \{p_{i0}, p_{i1}, ..., p_{iP}\}$, where $p_{ij}$ corresponds to the $j$-th passage with respect to the query term $u_i$.

The **interaction network** was designed to independently evaluate each query-passage interaction, producing a final relevance score per sentence. In more detail, it receives as input the **query**, $q$, and the **aggregated set of passages**, $D(u_i)$, and creates for each query-passage pair a similarity tensor (interaction matrix) $S \in [-1, 1]^{Q \times T}$, where each entry $S_{ij}$ corresponds to the cosine similarity between the embeddings of the $i$-th query term and $j$-th passage term, $S_{ij} = \frac{\vec{u_i}^T \cdot \vec{v_j}}{\|\vec{u_i}\| \times \|\vec{v_j}\|}$. Next, a 3 by 3 convolution followed by a concatenation of a the global max, average and average k-max polling operation are applied to each similarity tensor, to capture multiple local relevance signals from each features map, as described in Equation 1,

$$
\begin{aligned}
h_{i,j}^m &= \sum_{s=0}^{2} \sum_{t=0}^{2} w_{s,t}^m \times S_{i+s,j+t} + b^m \,, \\
h_{max}^m &= \max(h^m), \ m = 1, ..., M \,, \\
h_{avg}^m &= \text{avg}(h^m), \ m = 1, ..., M \,, \\
h_{avg-kmax}^m &= \text{avg(k-max}(h^m)), \ m = 1, ..., M \,, \\
h &= \{h_{max}; h_{avg}; h_{avg-kmax}\}.
\end{aligned}
\tag{1}
$$

Here, $w$ and $b$ are trainable parameters, the symbol ';' represents the concatenation operator, $M$ corresponds to the total number of filters and the vector $\underset{3M \times 1}{\vec{h}}$ encodes the local relevance between each query-passage, extracted by these pooling operations. At this point, the aggregated set of passages, $D(u_i)$, is now represented by their respective vectors $\vec{h}$, i.e., $D(u_i) = \{\vec{h_{p_0}}, \vec{h_{p_1}}, ..., \vec{h_{p_P}}\}$.

The final step of the interaction network is to convert these passage representations $\vec{h}$ to a final relevance score, for which we employed a fully connected layer with sigmoid activation. Here the intuition is to derive a relevance score, relevant (1) or irrelevant (0), directly from the information that was extracted by the pooling operators. So, after this stage, the aggregated set of passages, $D(u_i)$, it is represented by this relevance score, i.e., $D(u_i) = \{s_{p_0}, s_{p_1}, ..., s_{p_P}\}$ or $D(u_i) = \vec{s_{u_i}}$.

Next, the **aggregation network** takes into consideration the importance of each query term by using a gating mechanism, similar to the DRMM (Guo et al., 2016), over the aggregated set of passages as described in Equation 2, i.e., each passage score is weighted according to the importance of its associated query term.

$$c_{u_i} = \underset{1 \times E}{\vec{w}} \cdot \underset{E \times 1}{\vec{x_{u_i}}} ,$$
$$a_{u_i} = \frac{e^{c_{u_i}}}{\sum_{u_k \in q} e^{c_{u_k}}} , \qquad (2)$$
$$\underset{P \times 1}{\vec{s_{u_i}}} = \underset{1 \times 1}{a_{u_i}} \times \underset{P \times 1}{\vec{s_{u_i}}} .$$

Here, $w$ is a trainable parameter and $\vec{x_{u_i}}$ corresponds to the embedding vector of the $u_i$ query term. Then the distribution of each query term importance, $a$, is computed as a softmax and applied to the respective passages scores.

To produce the final document score a scorable vector, $\vec{s}$, is created by performing a summation alongside the query-terms dimension of $\vec{s_{u_i}}$. Note that in this step we could have explored other ways to produce this final vector, however, this approach seems to empirically work. Finally, this final scorable vector, $\vec{s}$, is fed to a Multi-Layer Percepreton (MLP) to produce the final ranking score.

Another interesting feature of this model relies on the architecture that is capable of providing relevance scores for each passage, $\vec{s_{u_i}}$, which can be explored to easily extract the passages that the model considered as more relevant, i.e., that most contributed to the final document score.

At last, it is noteworthy that this light architecture results in a model with a very low number of parameters. In the case of the Covid-19 dataset, the final model has only 620 trainable parameters. For training, adopted a cross-entropy pairwise loss, similarly to (Hui et al., 2018), with adam optimizer.

## 2.4 Adaptation to the Covid novel coronavirus literature and training details

As already mentioned, this system is an adaptation of our current working system for the BioASQ challenge. Furthermore, our motivation was that the knowledge learned from the BioASQ data could be transferred to this specific domain since a great similarity exists, in terms of documents, between both collections. For example, the BioASQ searches over the annual Pubmed Baseline, while a majority of the Cord-19 documents are from PubMed Central (PMC) full-text collection, according to (Wang et al., 2020), sharing the same Pubmed abstracts. More concretely, in the beginning we only trained the word2vec embeddings on the Pubmed baseline plus the Cord-19 collection (Pubmed+Cord-19) to accommodate new terms that didn't appear on the BioASQ collection, e.g., "Covid".

The $k1$ and $b$ parameters of BM25 were fine-tuned on the bioASQ data and the neural ranking model was also trained on the BioASQ data using the Pubmed+Cord-19 embeddings.

## 3 Evaluation

TREC-Covid is a challenge launched by AI2 and the National Institute of Standards and Technology (NIST) to the information retrieval and text processing communities, to create systems capable of searching the growing literature related to the novel coronavirus. The challenge follows the TREC conventions and is divided into five individual rounds, where each team should submit a ranking order of relevant documents, retrieved from the CORD-19 collection, for each topic. A topic is a description of the information need, in this case, it is composed of three fields: a query, i.e., most relevant terms; a question, i.e., natural language question; and a narrative, i.e., a more detailed description of the information need with respects to the documents. At least one submission per team is then manually evaluated by domain experts and the union of every evaluation contributes to the TREC-Covid relevant feedback dataset.

Given this opportunity, we decided to put our system to the test in this challenge, both as a way to produce empirical measures of performance and also to continually improve the current solution. At the time of writing, this challenge is currently on the fourth round, which means that only the results of the first, second and third rounds are available.

An important note is that, since CORD-19 is a continually growing collection, each round uses a different snapshot. The task for the first round was to retrieve documents for a total of 30 topics with no feedback data available to use. Each subsequent round then re-used the topics from previous rounds, for which feedback data was also made available, plus five new topics. Additionally, evaluation was performed in residual manner, which means that the documents evaluated in previous rounds were discarded from the evaluation of subsequent rounds. Systems were measured in terms of nDCG@10, P@5, bpred and Map, with nDCG@10 being the adopted ranking metric.

Each team participating in the challenge could submit a maximum of three runs, which could be either a manual run, if some manual action was performed after looking at the topics, a feedback run, if the available feedback data was used, or automatic.

## 4 Results and Discussion

In this section we show and discuss our results for each round separately, since we utilize the feedback from the previous round to continuously improve our solution. For each round, we present the Top 3 performing systems and our best submission comparatively. Additionally we also compare it with the ranking order before reranking, i.e. the output from phase-I, represented by the name **baseline**.

### 4.1 Round 1

The first round received a total of 143 runs from 56 teams, of which 100 runs were automatic, according to (Voorhees et al., 2020). Our top run (BioinformaticsUA) consists of the previously described pipeline finetuned and trained on the BioASQ with the PubMed+Cord-19 embeddings and using the **question** field of the topics as the information need. Table 1 presents a comparison with the three top performing systems and a more complete leaderboard is available here https://tinyurl.com/trec-covid-rnd1.

Given the highly competitive submissions, our

| System | nDCG@10 | P@5 |
|---|---|---|
| sab20.1.meta.docs | **0.6080** | **0.7800** |
| SLEDGE (MacAvaney et al., 2020) | 0.6032 | 0.6867 |
| IRIT_marked_base | 0.5880 | 0.7200 |
| **Ours** | 0.5298 | 0.6333 |
| Phase-I baseline | 0.4633 | 0.5933 |

Table 1: Comparison to the top 3 automatic runs in terms of nDCG@10 and P@5 against our best submission and our reranking baseline for the first round.

simple pipeline was able to achieve competitive results, being ranked at position 12 of 100 runs. These were compelling results, especially when compared with the second and third top runs that, respectively, explored SciBERT (Beltagy et al., 2019) and BERT (Devlin et al., 2018) as reranking alternatives, which are much more expensive models and ultimately more expensive to serve for public use. Additionally, our system was able to outperform other runs that were also based on BERT and T5 (Raffel et al., 2019), which reinforces our original point on the challenge of finetuning these very large models.

On the other side of the spectrum, the best run used a more traditional technique based on the Vector Space Model with lnu.ltu weighting. Notably, other simplistic approaches that explored BM25, query rearranging and expansion, and/or pseudo-relevance feedback, also achieved remarkable performance. This may be explained by the lack of training data, which may injure most of the neural ranking approaches in terms of generalization to the new domain.

The last line on Table 1 presents the evaluation of the ranking order before reranking, which shows that our reranking model was capable of successfully using the context to refine the exact signal matches captured by BM25. Furthermore, we also considered that our solution in phase-I is underperforming compared to similar alternatives presented by other teams.

### 4.1.1 Lessons learned

As a quick overview, we now address the notable points learned that have been used to prepare the second round. Firstly, BM25 should be able to produce a better ranking order by enhancing the query with more relevant terms as suggested in the "udel_fang_run3" run. Also, the ensemble of multiple runs proved to be beneficial, particularly the reciprocal rank fusion (Cormack et al., 2009), which was widely adopted during the competition.

## 4.2 Round 2

For the second round, based on the insights gathered from the previous round, we decided to use the "udel_fang_run3" queries for phase-I, BM25 search. Then for phase-II we employed an ensemble of five runs of our neural ranking model pretrained on the BioASQ data and finetuned following a 5-fold setup over the feedback data from the first round. The TREC organizers, for this round, received a total of 136 runs from 51 teams. Table 2, similarity to the previous, shows a comparison against the top three performing runs, with a more complete view available here `https://tinyurl.com/trec-covid-rnd2`. Additionally, the last two lines represent the TREC baselines that correspond to the ensemble of multiple runs obtained using the Anserini system[7].

| System | nDCG@10 | P@5 |
|---|---|---|
| SparseDenseSciBert | **0.6772** | 0.7600 |
| mpiid5_run1 | 0.6677 | **0.7771** |
| UIowaS_Run3 | 0.6382 | 0.7657 |
| **Ours** | 0.5016 | 0.5943 |
| Phase-I baseline | 0.2623 | 0.3029 |
| TREC Baseline (r2.fusion2) | 0.5553 | 0.6800 |
| TREC Baseline (r2.fusion1) | 0.4827 | 0.6114 |

Table 2: Comparison to the top 3 feedback runs in terms of nDCG@10 and P@5 against our best submission and our reranking baseline.

From the table of results, it is observable that our system had poor performance when compared with other runs. In more detail, the major drawback occurs in the baseline before the reranking, where it unperformed in both metrics by a factor of two when comparing with the TREC baseline. These inconsistent results seem to have been caused by an undetected error, since the methodology used to create the TREC Baseline is quite similar to our adopted changes in phase-I. Additionally, even though the baseline underperformed, the reranking was still capable of considerably boosting the original ranking order, which also reinforces the idea of some inconsistency. It is worth recalling that this evaluation is performed in a residual manner, since it uses the first 30 topics from the first round, which means that the previous feedback for the first 30 topics do not account for this evaluation.

Concerning the other systems, the top-performing among the feedback runs used SciBert

as a reranker, while the second relied on the ELECTRA (Clark et al., 2020) model, also another transform-based model. Interestingly the third top performing run, from the UIowaS team, used a more traditional approach, employing an ensemble of two runs produced with relevance feedback for the first 30 topics, using the feedback from the first round, and pseudo-relevance feedback for the new topics.

### 4.2.1 Lessons learned

From the descriptions of the top runs, we observed that the majority of the teams that used neural ranking solutions adopted SciBert, although other transform-based alternatives were also used, such as ELECTRA, Bert, and T5. Additionally, the number of top runs that explored these neural ranking models seemed to increase, which may be related to the availability of training data from the first round. Another interesting detail concerns the number of documents that are ranked. Up to this round, we reranked the top-1000 documents retrieved during phase-I. However, if using a stronger baseline built on additional signals beyond the exact match, such as relevance feedback, it may be beneficial to rerank a smaller subset, with the intuition of preserving some of the original order and apply reranking to less extent.

We also would like to reinforce the importance of relevance feedback techniques for building a runs for this particular challenge, given the nature of the residual evaluation. In other words, this evaluation offers an excellent setting to use relevance feedback techniques, since it has available a large amount of relevant documents to the majority of the topics that belong to the test set for each round.

## 4.3 Round 3

Building upon the information gathered from the previous rounds, we opted to use the UIowaS baseline, which was kindly shared with the community, as our baseline, i.e., it replaces the phase-I step. Phase-II consisted of the same neural ranking model presented in this paper, pretrained on the BioASQ data, and finetuned with feedback data from rounds 1 and 2. Furthermore, we chose only to rank the top-K documents of the baseline. To find the value of K we performed experiences with the round 2 model and the respective UIowaS baseline, selecting $K = 10$ as an optimal value for boosting both metrics.

---

[7]https://github.com/castorini/anserini/blob/master/docs/experiments-covid.md

| System | nDCG@10 | P@5 |
|--------|---------|-----|
| covidex.r3.t5_lr | **0.7740** | 0.8600 |
| **Ours** | 0.7715 | 0.8650 |
| UIowaS_Rd3Borda | 0.7658 | **0.8900** |
| Phase-I baseline (UIowaS) | 0.7617 | 0.8750 |
| TREC Baseline (r3.rf) | 0.6883 | 0.7950 |
| TREC Baseline (r3.fusion2) | 0.6100 | 0.7150 |

Table 3: The top 3 runs (overall) in terms of nDCG@10 and P@5. Followed by the adopted baseline and both TREC baselines.

The third round received a total of 79 runs from 31 teams. Results are presented in Table 3 and a more complete leaderboard is available in `https://tinyurl.com/trec-covid-rnd3`.

In this round, our top submission ranked second amongst 79 runs. When compared against the baseline, it is possible to notice that there was only a small improvements in terms of nDCG@10, which may make the quality of the reranking questionable. However, both metrics are already quite high, arguably, making the comparison less discriminative. Worth of mention is another run from the UIowaS team that tried to rerank their baseline but failed to obtain an improvement, which highlights the presented reranking challenge.

Regarding the other runs, the top-performing run also explored an insightful strategy, that consists of interpolation between the scores provided by a finetuned T5 model and a logistic regression classifier trained on tf-idf from round 1 and 2. The third best run belongs to the UIowaS team and shares the same principle described in Section 4.2. According to the run description[8], this should correspond to our phase-I baseline results, despite the different results shown in the table. We believe that this difference can be explained by a reported bug that caused some judgments to be missing from the feedback data that we used to measure the baseline performance.

### 4.3.1 Lessons learned

It is clear that the "heavy lifting" is done by the strong baseline from the UIowaS team and this run just tries to explore this strong baseline by improving the top documents, i.e., focusing more on top precision. Nonetheless, it is also noteworthy that the neural ranking model that explores a different set of relevance signals was able to indeed boost the original baseline. Furthermore, this also empiri-

cally reinforces the idea of only reranking a smaller subset of documents. However, a better approach to make this combination is a clear route to explore.

## 5 Performance comparison

In this section, we empirically analyse the performance of our neural ranking solution by comparing it against several transform based-models.

Following the literature, we selected BERT (Devlin et al., 2018) model and BERT variants, since they seem to be the most widely adopted. More precisely in our tests we use the following pretrained models:

- BERT (x12) (Devlin et al., 2018) with 12 layers and 110 million parameters

- BERT (x24) (Devlin et al., 2018) with 24 layers and 340 million parameters

- distilBERT (Sanh et al., 2020) with 6 layers and 66 million parameters

- ALBERT (Lan et al., 2020) with 12 layers and 11 million parameters

Regarding the methodology, we performed an inference time evaluation over 100 queries, each associated with 250 documents, totalling 25000 query-documents pairs as our samples. Furthermore, we performed all the tests in the machine presented on Table 4 with TensorFlow 2.2.0 and CUDA 10.1, and decorated the models with the "tf.function"[9] converting them into a static computation graph for performance reasons. For the transform-based models we used the HuggingFaces (Wolf et al., 2019) library with the corresponding TensorFlow version. Following the literature (Yang et al., 2019; Nogueira et al., 2019; Dai and Callan, 2019), we fed to the transform-based models the input "[CLS] query tokens [SEP] document tokens [SEP]" and further fed the resulting "[CLS]" representations to a multi layer perception computing the final document score.

Table 4: Specification of the hardware used during the inference times experiments.

| CPU | 2x Intel(R) Xeon(R) CPU E5-2630 v4 @ 2.20GHz |
|-----|-----|
| GPU | Nvidia Tesla K80 with 12 GB |
| RAM | 128 GB |

---

[8]https://ir.nist.gov/covidSubmit/archive/round3/UIowaS_Rd3Borda.pdf

[9]https://www.tensorflow.org/versions/r2.2/api_docs/python/tf/function

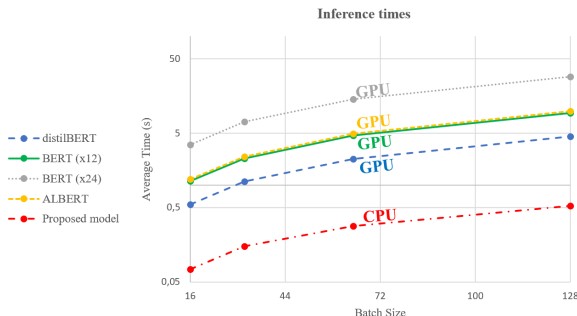

Figure 4: Comparison of the inference times of the proposed model running on CPU against several transform-based models running on GPU.

Figure 4 shows a comparison of the inference times, measured in seconds, of our proposed model against the previously enumerated transform-based alternatives, for varying batch sizes. Our model was 7.5 faster than distilBERT, the fastest transform-based model variant, and 212 times faster than BERT (x24). Significantly, our model was able to perform inference in less than one second, for a batch size of 128 and while running on CPU, which makes it more suitable for deploying in real-world search applications.

## 6 Conclusion

This paper presents and analyses several adaptations of an information retrieval pipeline to address the challenge of Covid-19 literature search. Our system is based on previous work on the BioASQ 8b challenge that we adapt to the novel coronavirus literature. The system follows a two-stage retrieval pipeline, with BM25 as the first stage and a light interaction-based model, with only 620 trainable parameters, as the second stage.

We evaluate our system performance on the three available rounds of the ongoing TREC Covid competition, where in general we obtained competitive results against a wide variety of solution. We also explored the trends of the top submissions to evolve our initial approach. In terms of results, our best ranks were twelfth in one hundred, in round 1, and second in seventy-nine in round 3.

As more direct future work we propose to continually explore a better reranking strategy to apply on a strong baseline, especially a baseline based on relevance feedback that seems to be suitable to this residual evaluation setting. Another line of work could encompass combining a transform-based model, like BERT, in our architecture, since these are well suited to our objective of better evaluating the passage context. This combination may be achieved by simply replacing the word2vec embeddings by context embeddings or by completely replacing the interaction network.

## Acknowledgments

This work has received support from the EU/EFPIA Innovative Medicines Initiative 2 Joint Undertaking under grant agreement No 806968 and from National Funds through the FCT - Foundation for Science and Technology, in the context of the project UIDB/00127/2020.

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
