# OpenReview forum: "Frugal neural reranking: evaluation on the Covid-19 literature"
_EMNLP/2020/Workshop/NLP-COVID — NLP-COVID19-EMNLP Oral_

### Official Review · AnonReviewer1 · 2020-09-24
**A solid contribution with a usable prototype implementation**

**Rating:** 8
**Confidence:** 3

**Review:**

**Core review**:

This paper presents a currently in-development system for information retrieval in the CORD-19. Authors present a pipeline that is being evaluated in the TREC-COVID challenge and shows a clear advantage with respect to other approaches despite (or thanks to) being based on a simpler neural ranking component. While it may not necessarily be the top-performing approach, authors claim that the simplicity of the neural ranking component and the overall pipeline makes it more feasible to deploy online and use for real-time queries. However, the experimental data provided in the paper (which I briefly cross-checked w.r.t. the TREC-COVID leaderboard links provided in the paper) positions this system with a very good performance, among the top-3 in several metrics. A prototype of the system is deployed and usable online (I've been able to perform some queries while others have taken too long).

In general, I believe this paper provides a valuable approach and a tool the community can benefit from, and I think it is very relevant for this workshop. My only suggestion is regarding the evaluation. As it is currently written in the paper, there is too much emphasis (in my opinion) on the idiosyncrasies of the TREC-COVID challenge, which makes the evaluation section unnecessarily long. On the other hand, I would appreciate some experimental results to back the hypothesis that this type of system is easier to deploy and faster to execute than the alternatives.

**Reasons to accept:** The paper presents a clear result with demonstrable effectiveness, as validated by experimental data that is publicly available and a prototype system available online.

**Reasons to reject:** Some claims regarding the scalability and practical advantages of this system require further experimentation to be completely justified.

---

> ### Author Response · Authors · 2020-09-27
> **Thank you for your comments but unfortunately we don’t know yet if we will be able to run the additional experimental results in time.**
>
> We thank the reviewer for their positive comments as well as suggestions for improvement.
>
> We agree with the reviewer that the ‘Evaluation’ section could be more succinct in the final version of the paper. However, we feel that a description of the topics and some details on the rounds and residual evaluation would remain necessary.
>
> Regarding the experiments to support the practical advantages of such a lightweight system, unfortunately we do not believe we will be able to perform them in time for this revision. We however support our claims on the immense difference in number of parameters of our model (order of hundreds) to transformer based models (order of millions), which then impacts the number of operations required.

---

### Official Review · AnonReviewer2 · 2020-09-25
**A simple light-weight but strong performing reranking model**

**Rating:** 7
**Confidence:** 3

**Review:**

This paper proposes a very light-weight reranking model for retrieving documents from the CORD-19 dataset. The impact is that the proposed model would encourage wider application in cases where computation resources are a constraint. The proposed model, while only having 620 parameters, performed surprisingly well (ranked 2nd in the 3rd round).

The paper is well-written and well-motivated. I specially like the 3-round of testing and corresponding "lessons learnt" parts. It offers some insights into the technical details that would cause substantial difference in the evaluation performances, and thus inspire future works.

One limitation is the lack of proper explanation on the baseline in the 3rd round. The paper mentioned a bug in feedback data but still not sure why the baseline performance is missing. Although it said "... it should correspond to our phase-I baseline results", but where are the numbers? How far is it off from the system "ours"?

---

> ### Author Response · Authors · 2020-09-27
> **Thank you for your comments, the "UIowaS baseline (phase-I)" corresponds to "our baseline"**
>
> We thank the reviewer for their positive comments as well as suggestions for improvement.
>
> Our baseline results for round 3 are shown in Table 3 in the line marked as “UIowaS baseline (phase-I)”.
> The values in this line should be exactly the same as the line above (“UIowaSRd3Borda”), since we used that baseline run, which was shared by UIowaS for all participants, as our Phase-I baseline; that is, we applied our re-ranker to that run.
> As we mention in the paper, we believe that the difference can be explained by a reported bug that caused some judgments to be missing from the feedback data that we used to measure the baseline performance.
>
> We agree that the text describing these results is slightly confusing and can be made clearer. We also propose changing this table entry to “Phase-I baseline (UIowaS)” to make it clearer, and change the corresponding entries in Tables 1 and 2 to “Phase-I baseline”.

---

### Official Review · AnonReviewer3 · 2020-09-26
**Light weight and efficient query-document linking algorithm with nice performance.**

**Rating:** 8
**Confidence:** 3

**Review:**


This work introduces a light weight retrieval and reranking model, as well as an easy-to-use web search service. Although the model has only 620 parameters and does not use the sophisticated Transformer variants like Bert,  it still ranks the second position in the third round of submission.

The writing quality of the paper is pretty good, and the logic is nice. I believe that the work is innovative.

There are still several weak points worth mentioning.

1. It seems that the preliminary work of the model  come from (Almeida and Matos. In 2020). This work should be cited at the first moment when mentioned. Say, the third paragraph of the introduction, "BioASQ system in our work".

2. Please add citation to “BM25” in the section 2.2.

3. Are both the query and paragraph initialized by a pre-trained Word2vec model? There is no clear explanation in the experiment part, but the readers could get the above clue in the last paragraph. So, if "yes", declare them clearly earlier, pls.

4. The effect of the re-ordering model on the third round of Baseline is weak, so does the UIowaS team. That's interesting. Any more analysis?

---

> ### Author Response · Authors · 2020-09-27
> **Thank you for your comments, we have addressed all the suggested changes and the final re-ordering behaviour**
>
> We thank the reviewer for their positive comments as well as suggestions for improvement.
>
> We agree with points 1 and 2.
>
> The following sentence is missing in section 2.3, to clearly indicate the use of a pre-trained word2vec model. We thank the reviewer for pointing this out.
> “A pre-trained word2vec model is used for representing query and document terms.”
>
> Regarding the results of round 3, we didn't further explore these results, but this apparent weak improvement may be related to the poor discrimitave capabilities of the evaluation metrics (P@5 and NDCG@10) at this stage of the competition. Also, the scores provided by the baseline are pretty high, which leave “less room for improvement” since we are only retaking the top-10 as mentioned on the paper. This is also verified by the other submission from the UIowaS team that tries to rerank the same baseline, but failed to achieve an improvement. Their reranking approach achieved NDCG@10: 0.7175 and P@5 0.8000 and can be examined here: https://ir.nist.gov/covidSubmit/archive/round3/UIowaS_Rd3MLReRank.pdf